# Predicting Blast-Induced Ground Vibration in Open-Pit Mines Using Vibration Sensors and Support Vector Regression-Based Optimization Algorithms

**DOI:** 10.3390/s20010132

**Published:** 2019-12-24

**Authors:** Hoang Nguyen, Yosoon Choi, Xuan-Nam Bui, Trung Nguyen-Thoi

**Affiliations:** 1Institute of Research and Development, Duy Tan University, Da Nang 550000, Vietnam; 2Department of Energy Resources Engineering, Pukyong National University, Busan 48513, Korea; 3Department of Surface Mining, Mining Faculty, Hanoi University of Mining and Geology, 18 Pho Vien, Duc Thang ward, Bac Tu Liem district, Hanoi 100000, Vietnam; buixuannam@humg.edu.vn; 4Center for Mining, Electro-Mechanical Research, Hanoi University of Mining and Geology, 18 Pho Vien, Duc Thang ward, Bac Tu Liem district, Hanoi 100000, Vietnam; 5Division of Computational Mathematics and Engineering, Institute for Computational Science, Ton Duc Thang University, Ho Chi Minh City 700000, Vietnam; 6Faculty of Civil Engineering, Ton Duc Thang University, Ho Chi Minh City 700000, Vietnam

**Keywords:** peak particle velocity, vibration sensor, soft computing, evolutionary algorithm, hybrid model, open-pit mine

## Abstract

In this study, vibration sensors were used to measure blast-induced ground vibration (PPV). Different evolutionary algorithms were assessed for predicting PPV, including the particle swarm optimization (PSO) algorithm, genetic algorithm (GA), imperialist competitive algorithm (ICA), and artificial bee colony (ABC). These evolutionary algorithms were used to optimize the support vector regression (SVR) model. They were abbreviated as the PSO-SVR, GA-SVR, ICA-SVR, and ABC-SVR models. For each evolutionary algorithm, three forms of kernel function, linear (L), radial basis function (RBF), and polynomial (P), were investigated and developed. In total, 12 new hybrid models were developed for predicting PPV in this study, named ABC-SVR-P, ABC-SVR-L, ABC-SVR-RBF, PSO-SVR-P, PSO-SVR-L, PSO-SVR-RBF, ICA-SVR-P, ICA-SVR-L, ICA-SVR-RBF, GA-SVR-P, GA-SVR-L and GA-SVR-RBF. There were 125 blasting results gathered and analyzed at a limestone quarry in Vietnam. Statistical criteria like R^2^, RMSE, and MAE were used to compare and evaluate the developed models. Ranking and color intensity methods were also applied to enable a more complete evaluation. The results revealed that GA was the most dominant evolutionary algorithm for the current problem when combined with the SVR model. The RBF was confirmed as the best kernel function for the GA-SVR model. The GA-SVR-RBF model was proposed as the best technique for PPV estimation.

## 1. Introduction

Construction materials and energy are in great demand in every country, especially developing ones. As a result of this demand, opencast mines and quarries are flourishing and displaying high levels of productivity to meet market requirements. Owing to this, ultimate pit boundaries are reached quickly, and this significantly affects neighboring residential areas and surrounding structures. Among the activities conducted in opencast mines, blasting is a necessary step which leads to serious environmental impacts, such as air and ground vibrations, fly-rock, noise pollution, back-break, and release of gases [1]. Of these harmful effects, ground vibration is considered to be the most dangerous phenomenon. It can make vibrations of buildings, instability of slopes and benches, and cause bewilderment for the residentials surrounding the mine.

To measure ground vibration induced by blasting operations in open-pit mine, peak particle velocity (PPV) is considered a standard criterion for evaluating the intensity of the ground vibration. More than 80% of the energy of explosives is released by generating ground vibrations that spread through the rock and soil to surrounding structures [2,3]. With a large oscillation amplitude over a short period, blast-induced PPV significantly affects these surrounding structures. More specifically, in the case of blast-induced PPV oscillations that concur with the natural vibrations of the building, resonance phenomena happen, which can cause substantial structural damage, crashes or cracks, and instability of the bench and slope in opencast mines [4,5]. Many households around blasting areas feel disconcerted by the effects of blast-induced ground vibrations. Complaints and litigation related to blast-induced PPV are serious issues, causing many opencast mines to stop production. Therefore, accurate prediction of blast-induced PPV and control of harmful effects caused by the blasting operations are significant challenges for open-pit mines. Precise forecast models of blast-induced PPV are necessary to minimize unwanted injuries in the surrounding environment.

In this regard, artificial intelligence (AI) applications are considered useful, not only as robust techniques in the mining field but also in many other areas (e.g., civil engineering, fuel, and energy, and environment) [2,4,6,7,8,9,10,11,12,13,14,15,16,17,18,19,20,21,22,23,24]. An overview of the literature related to PPV prediction showed that many AI models have been developed and proposed, as listed in Table 1. The reports in Table 1 showed that the ANN (artificial neural network) seem to be the most common approach for predicting ground vibration. The enhance algorithm of ANN (i.e., Levenberg–Marquardt algorithm) was also applied to improve the accuracy of the ANN model in predicting ground vibration. Besides, the hybrid of ANN and optimization algorithms (e.g., the particle swarm optimization (PSO), imperialist competitive algorithm (ICA), artificial bee colony (ABC)) was also studied and proposed—aiming to provide high reliability in predicting blast-induced ground vibration. Although the proposed models by the previous researchers are outstanding, they were not applied in other areas/regions/countries. Furthermore, new hybrid models with better accuracy are always the goal of researchers, especially benchmark models-based approach. Of those, support vector machine for regression problems (SVR) is considered as one of the common benchmark models, which was applied in many fields [25,26,27,28,29,30,31]. Our best review of the related works showed that hybrid models based on SVR and evolutionary algorithms were rare. The SVR model does not seem to be considered for optimization by evolutionary algorithms in predicting PPV. Only the PSO-SVR model has been developed to predict blasting issues, although the expected object is air over-pressure [32]. Moreover, kernel functions have been recommended that significantly affect the performance of SVR models, and these have been reviewed by Nguyen [33]. However, they have not been considered and evaluated when combined with evolutionary algorithms. Therefore, this study aims to assess the overall performance of different evolutionary algorithms such as PSO algorithm, GA, ICA, and ABC when they are combined with the SVR model using linear (L), radial basis function (RBF), and polynomial (P) kernel functions. 12 new hybrid models were developed, named the ABC-SVR-P, ABC-SVR-L, ABC-SVR-RBF, PSO-SVR-P, PSO-SVR-L, PSO-SVR-RBF, ICA-SVR-P, ICA-SVR-L, ICA-SVR-RBF, GA-SVR-P, GA-SVR-L, GA-SVR-RBF models. A comprehensive comparison and assessment of these models will be presented in this study.

## 2. Proposing the Framework of SVR-based Evolution Algorithms

As introduced above, this study aims to investigate and developed several hybrid models based on SVR and evolution algorithms, such as PSO, ICA, GA and ABC. Due to the details of PSO, ICA, GA, ABC, and SVR being introduced in many previous papers [47,48,49,50,51,52,53,54,55,56,57,58,59,60]; therefore, they were not introduced in this study. This study focuses on proposing novel hybrid framework of evolution algorithms (EAs) and SVR, called EAs-SVR framework. For the development of the SVR model herein, three forms of kernel functions were applied, including linear function (L), polynomial function (P), and radial basis function (RBF). They can be formulated as follow:

- Linear:(1)$$G\left(x,x_{i}\right)=x\cdot x_{i}$$

- Polynomial:(2)$$G\left(x,x_{i}\right)={\left[(x\cdot x_{i})+1\right]}^{d}\;\;\;;\;d=(1,2,\ldots )$$

- Radial basis function:(3)$$G\left(x,x_{i}\right)=\exp[\frac{-{\left\|x-x_{i}\right\|}^{2}}{\sigma^{2}}]$$

For each kernel function, there will be one or more hyper-parameters(s) adjusted to get the highest performance for the models. In this study, the hyper-parameter(s) of the SVR models was searched and selected by the EAs (i.e., PSO, ICA, ABC, GA) with the aim to establish optimal performance.

In this part, the framework of the EAs-SVR models is proposed and described. As stated above, PSO, ICA, ABC, and GA were applied to optimize the SVR’s parameters. So, the EAs-SVR models consist of PSO-SVR, ABC-SVR, ICA-SVR, and GA-SVR models. During the development of the EAs-SVR models, L, P, and RBF kernel function were applied. The SVR’s parameters with different functions of the kernel are shown in Table 2. Based on the initial parameters in Table 2, the ABC, ICA, PSO, and GA algorithms perform a searching procedure for the optimal values of SVR’s parameters. Root-mean-squared error (RMSE) was used as a fitness function for all EAs-SVR models during the development of the models, and it was calculated using Equation (4). For each best hybrid model obtained, the RMSE value is lowest respectively to the best-obtained SVR’s parameters. The loops for the optimization process are used to find the optimal hyper-parameters. Eventually, the final EAs-SVR models (i.e., ABC-SVR, ICA-SVR, PSO-SVR, GA-SVR) are defined. The framework of the EAs-SVR models for predicting PPV is proposed in Figure 1.

## 3. Statistical Criteria

For evaluating the quality, as well as the reliability level of the mentioned models, three indicators of performance were used, including RMSE, MAE, and R^2^.
(4)$$\mathrm{RMSE}=\sqrt{\frac{1}{n}\sum_{i=1}^{n}{\left(y_{i}-\hat{y_{i}}\right)}^{2}}$$
(5)$$R^{2}=1-\frac{\sum_{i}(y_{i}-{\hat{y}}_{i}{)}^{2}}{\sum_{i}{\left(y_{i}-\bar{y}\right)}^{2}}$$
(6)$$\mathrm{MAE}=\frac{1}{n}\sum_{i=1}^{n}|y_{i}-\hat{y_{i}|}$$
where n is the total number of data, $y_{i}$ is measured PPV, ${\hat{y}}_{i}$ is predicted PPV and $\bar{y}$ is mean of measured PPVs. 

## 4. Vibration Sensors and Experimental Datasets

A quarry in Vietnam was selected as a typical case. It is located within latitudes 20°25′40” N–20°26′20” N and longitudes 105°53′10” E–105°54′00” E (Figure 2). The area of the mine is ~0.86 Km^2^ with the production ~6 million tons/year. The geological conditions of this site were presented in [11]. Figure 3 shows the structure of geological of this site study.

For measuring the intensity of vibration at this mine, Micromate geophone sensors were utilized. Its structure consists of two parts: a magnet and a coil of wire. The magnet is suspended by a wire coil, as shown in Figure 4. A blast-induced ground vibration reading is shown in Figure 5. Finally, 125 PPV events were recorded by a Micromate vibration sensor (Instatel, Canada). Figure 6 illustrates the process of data collection and vibration sensor used in this study.

A review of literature showed that many factors affect PPV. However, it is zoned into two main groups, including a controllable parameters group and a uncontrollable parameters group [61]. The controllable parameters group is used by scientists to popularize predicting PPV due to the ability to collect data directly and accurately, including W (explosive charge per delay), R (monitoring distance), T (stemming), P (powder factor), the number of boreholes, B (burden), H (bench high), S (spacing) and time delay [4]. In this study, four controllable parameters were used as the input variables, namely W, R, B, and S, whereas, PPV was considered as the output factor. A hand-held GPS was used to measure R, namely X91B GPS receiver, as shown in Figure 7. It can achieve an accuracy in centimeter-level based on landmark coordinates built and adjusted in the local [2]. The other parameters (W, B, S) were exported from 125 blast patterns. Table 3 summarizes the database used in this research. The structure and distribution of the data are illustrated through the histogram of the datasets as shown in Figure 8.

## 5. Results and Discussion

Before developing the models, the dataset is needed to divide into two phases, including training and test phases. Accordingly, 80% of the whole dataset (101 blasting operations) was selected randomly as the training phase for the models’ development. The remaining 20% (24 blasting operations) was used to check the accuracy, as well as the reliability of the models developed based on the training dataset. To avoid over-fitting or under-fitting of the models, the dataset was scaled in the range of [−1, 1]. Besides, the Box-Cox transformation method [62] and 10-fold cross-validation technique [63] were also applied to transfer data and improve the accuracy of the models.

### 5.1. ABC-SVR Models

For the ABC-SVR models, the training dataset was used to develop the PPV predictive models. Three kernel functions were involved in developing the ABC-SVR models with various hyper-parameters, as shown in Table 2. The ABC algorithm implemented a global search procedure to define the optimal values of the SVR models through the framework in Figure 1. In the ABC algorithm, a trial and error procedure for the number of bees was conducted for the parameters to be optimized over, with the number of bees set equal to 100, 200, 300, 400, 500, respectively; RMSE was used as a fitness function to be minimized; The number of food sources to exploit for the bees was set to 50 and the limit of a food source to 100; The boundary of the parameters to be optimized was set as [–10;10]. The optimization process was repeated 1000 times to find the optimal RMSE value. After setting parameters for the ABC algorithm, the bee finds the optimal values for the SVR models with the corresponding parameters. The performance of the optimization process was illustrated in Figure 9, Figure 10 and Figure 11. The optimal results for the ABC-SVR models are shown in Table 4.

### 5.2. PSO-SVR Models

As in the ABC-SVR models, an initial SVR model was generated with different kernel functions. Subsequently, the SVR’s parameters were searched and optimized by the PSO algorithm. The best values of SVR’s parameters (after discovery by the PSO algorithm) were evaluated through RMSE as those used for the ABC-SVR models. The lowest RMSE is the optimal PSO-SVR model. To employ the global search procedure by the PSO algorithm, its parameters needed to be set up first, including:-The number of particle swarms (*p*);-The maximum particle’s velocity (*V_max_*);-The individual cognitive ($\phi_{1}$);-The group cognitive ($\phi_{2}$);-The inertia weight (*w*);-The maximum number of iteration (*m_i_*).

For this study, the PSO’s parameters were set as follow: *p* = 100, 150, 200, 250, 300; *V_max_* = 1.9; $\phi_{1}$ = $\phi_{2}$ = 1.6; *w* = 1.8, and *m_i_* = 1000. Once the PSO’s parameters were established, the particles searched in a constrained space to find out the best place (under lowest RMSE). The lowest RMSE, the optimal PSO-SVR model was defined, and its hyper-parameters were extracted to build the optimal model. Figure 12, Figure 13 and Figure 14 show the performance of the PSO-SVR models while searching optimal values. Table 5 lists the optimal parameters of the PSO-SVR models.

### 5.3. ICA-SVR Models

For optimization of the SVR model by the ICA, the ICA’s parameters were established as the first step. A trial and error procedure for the number of initial countries (*N_country_*) and initial imperialists (*N_imper_*) was applied with *N_country_* was set equal to 100, 150, 200, 250, 300, respectively; *N_imper_* was set equal to 10, 20, 30, respectively. The maximum number of iterations (*N_i_*) was set equal to 1000 to ensure the stability of the model; the lower-upper limit of the optimization region (*L*) was placed in the range of –10 to 10; the assimilation coefficient (*As*) was set equal to 3; the revolution of each country (*r*) was set equal to 0.5. After the parameters of the ICA were established, empires perform a competition to find out the most substantial empire where the SVR’s parameters are optimized. The ICA-SVR’s performances are shown in Figure 15, Figure 16 and Figure 17. Finally, the ICA-SVR’s parameters were listed in Table 6.

### 5.4. GA-SVR Models

Concerning the GA-SVR models, the similar steps as those conducted for the previous models was applied. An initial SVR model with kernel functions was established as the first step; then, the parameters of GA were set up as the second step (e.g., mutation probability (P_m_), crossover probability (P_c_), and the number variable (*n*), the number of populations (*p*)). In this study, *P_m_* was set equal to 0.1; *P_c_* was set equal to 0.9; *n* = 4, and *p* was set equal to 100, 150, 200, 250, 300, respectively. RMSE was used as the fitness function according to Equation (4). The maximum iteration number was repeated 1000 times to ensure finding out the best values of the GA-SVR models. Figure 18, Figure 19 and Figure 20 report the GA-SVR’s performances through the RMSE. Eventually, the optimal GA-SVR’s parameters were extracted in Table 7.

### 5.5. Evaluating the Developed Models

Once the optimal parameters of the 12 hybrid models were achieved, 24 blasting events in the set of testing data were used to confirm the developed models’ accuracy. Table 8 presents the results of the 12 hybrid models on the sets of data.

Observing the statistical criteria on both the sets of training and testing data, the performance of the proposed hybrid models was very good. The developed models seem to be very stable, and the differences between the training dataset and the testing dataset are limited. However, if one only looks at the numbers in Table 8, it is tough to evaluate which model is the best in the present study. Therefore, a ranking method and intensity of color was used for assessing the efficiency of the developed models. Whereas, the green color represents the highest performance, the white color represents the lowest performance. The highlight of the GA-SVR-RBF indicated that it was the best model. Next are the ABC-SVR-RBF, ABC-SVR-P, ICA-SVR-RBF, PSO-SVR-RBF, GA-SVR-P, PSO-SVR-P, ICA-SVR-P, GA-SVR-L, ABC-SVR-L, PSO-SVR-L, and the last is ICA-SVR-L. It is of interest to consider the accuracy of the ABC-SVR-RBF, ABC-SVR-P, and ICA-SVR-RBF models. Although the ABC-SVR-RBF model provided lower accuracy than the ABC-SVR-P and ICA-SVR-RBF models on the testing dataset, the total ranking of the ABC-SVR-RBF model was higher than these two above models. The main reason is due to the fact that ABC-SVR-RBF’s performance was higher than the ABC-SVR-P and ICA-SVR-RBF models on the training dataset. In other words, the ABC-SVR-RBF model yielded more stable results than the ABC-SVR-P and ICA-SVR-RBF models in terms of the PPV prediction in the present study. Remarkably, the RBF kernel function seems to bring higher levels of accuracy over the linear and polynomial kernel functions. In contrast, the linear kernel function provided the lowest performance for the models in predicting PPV. As per the results of this study, the non-linear relationship of the variables was clarified. Figure 21 interprets the accuracy of the 12 proposed hybrid models. Note that, the intensity of the green color represents for the accuracy of the models. Greater intensity of green color, greater performance.

### 5.6. Sensitivity Analysis

In this study, four evolutionary algorithms (i.e., ABC, ICA, PSO, and GA) have played an essential role in optimizing the SVR models. Of those developed hybrid models, the GA-SVR-RBF model provided the highest level of accuracy as well as performance. As stated above, the RBF model was considered as the most useful function for the development of the SVR-based models using four evolutionary algorithms. However, determining which input factor(s) are the most important with the GA-SVR-RBF model, is a complicated problem; therefore, a procedure to assess the importance of input variables has been carried out in this study based on the selected GA-SVR-RBF model. The Hilbert-Schmidt Independence Criterion (HSIC) method was employed to analyze the input variables importance [64,65]. The results showed that R and W are the most important factors for estimating PPV, as illustrated in Figure 22.

## 6. Conclusions

AI has become more common in all fields, especially in technical fields. Applications of AI have helped to improve technical issues, especially in the mining industry. In this study, 12 hybrid models based on four evolutionary algorithms and SVR model were developed and comprehensively assessed. Some conclusions were drawn as follow:(1)Evolutionary algorithms are of great value in improving the accuracy of traditional models for PPV estimation, particular the SVR model.(2)Kernel functions have a great effect on SVR’s accuracy, especially the RBF. By means of evolutionary algorithms, kernel functions can reach optimal values for the SVR model.(3)GA is the most dominant evolutionary algorithm when combined with the SVR model and RBF (i.e., GA-SVR-RBF model) for predicting PPV. It should be approved as a robust technique to accurately predict PPV.(4)Monitoring distance and explosive charge (per delay) are the most critical factors in predicting PPV. They should be given special attention and carefully collected to improve the models’ accuracy in practice.

## Figures and Tables

**Figure 1 sensors-20-00132-f001:**
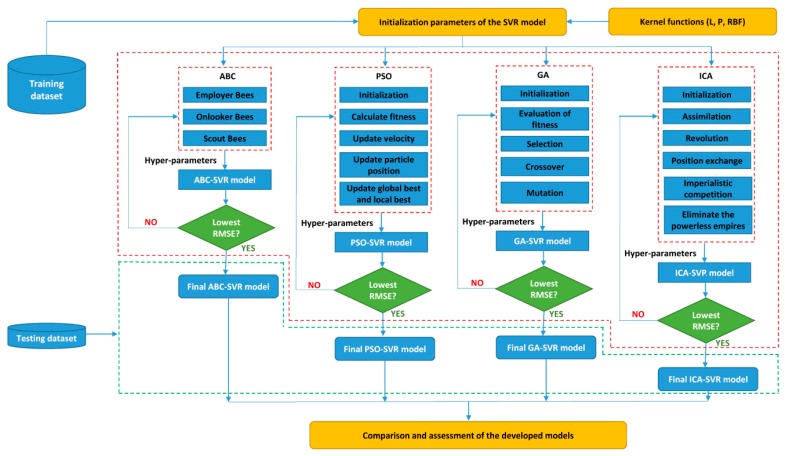
Framework of the support vector regression (SVR)-optimized by the four evolutionary algorithms for predicting peak particle velocity (PPV).

**Figure 2 sensors-20-00132-f002:**
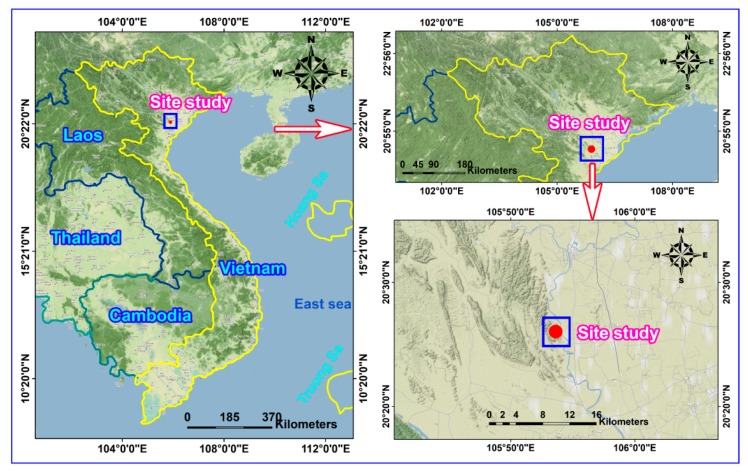
View of the site study.

**Figure 3 sensors-20-00132-f003:**
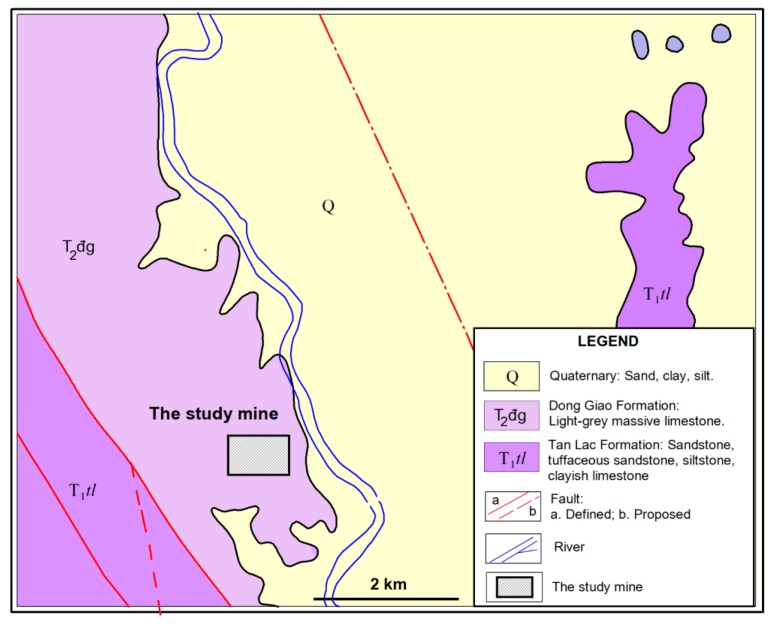
Structure of geology of the site study.

**Figure 4 sensors-20-00132-f004:**
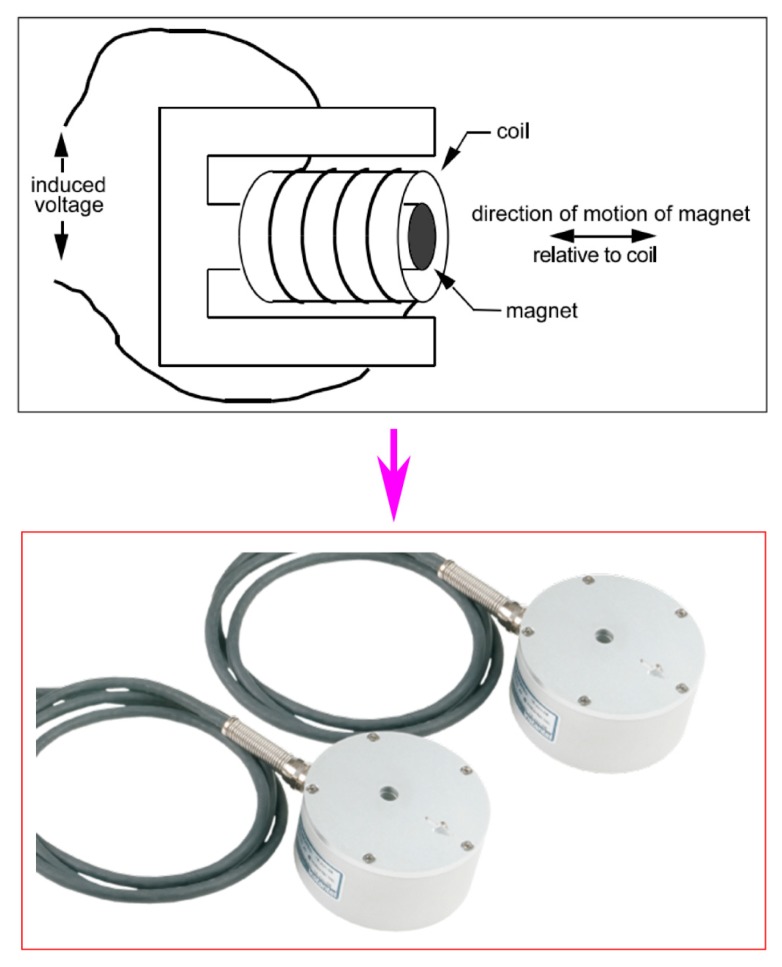
Structure of the geophone sensor for measuring vibration.

**Figure 5 sensors-20-00132-f005:**
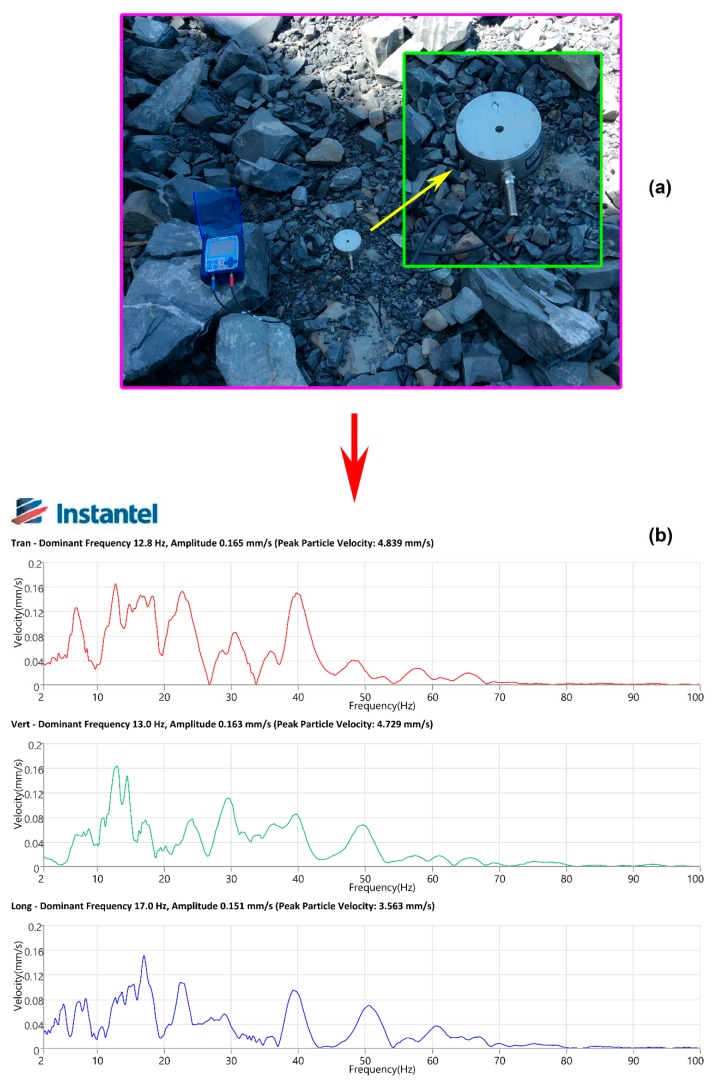
Vibration sensor and a result of the blast-induced ground vibration. (**a**) Micromate instrument (Instatel, Canada); (**b**) A result of vibration in open-pit mine.

**Figure 6 sensors-20-00132-f006:**
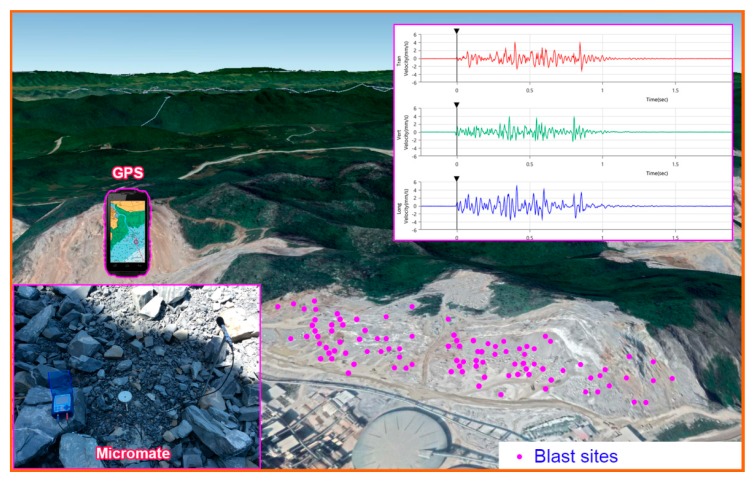
Data collection and a result of the PPV.

**Figure 7 sensors-20-00132-f007:**
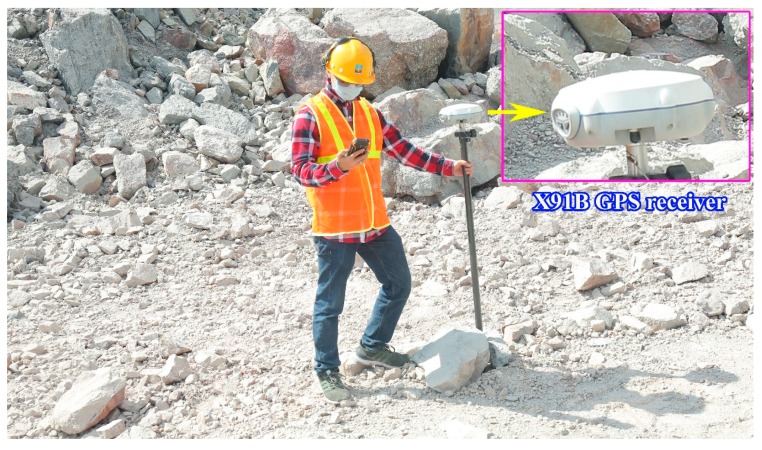
X91B GPS receiver used for measuring the distance in this study.

**Figure 8 sensors-20-00132-f008:**
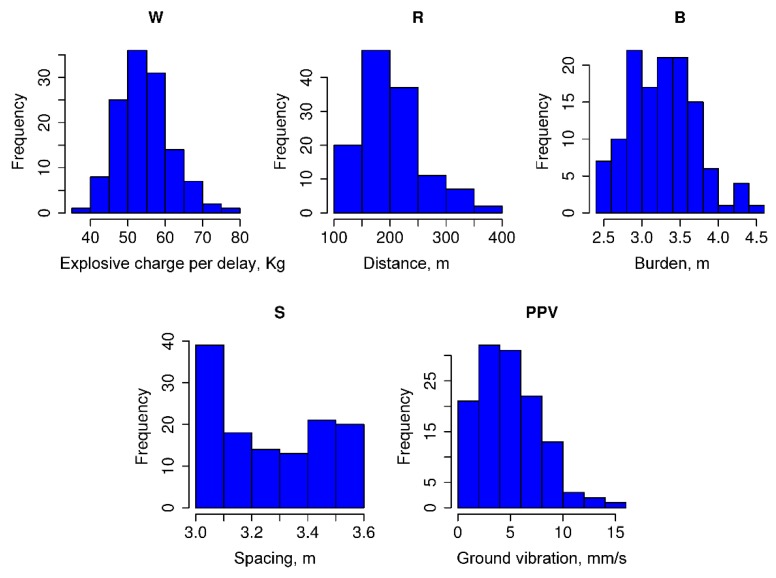
Histograms of the dataset used in this study.

**Figure 9 sensors-20-00132-f009:**
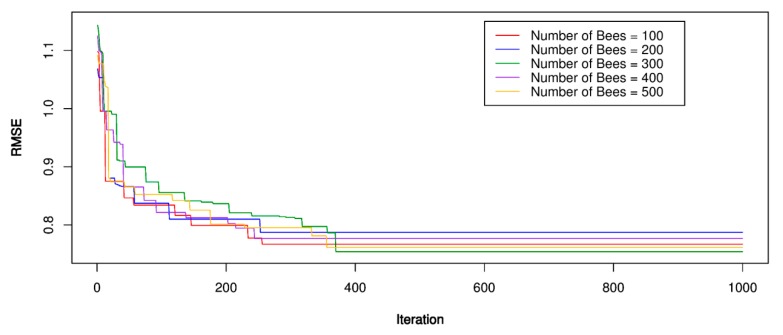
RMSE of the proposed artificial bee colony (ABC)-SVR-L model.

**Figure 10 sensors-20-00132-f010:**
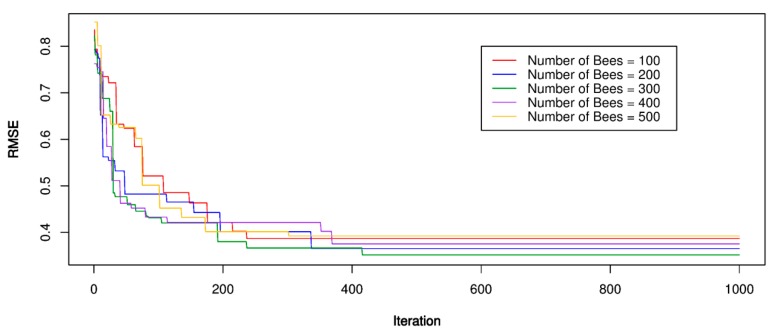
RMSE of the proposed ABC-SVR-P model.

**Figure 11 sensors-20-00132-f011:**
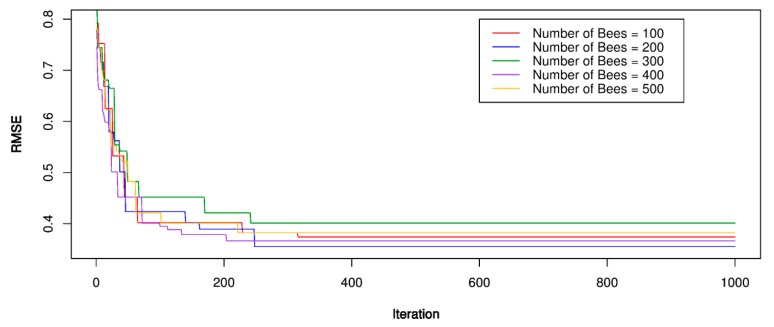
RMSE of the proposed ABC-SVR-RBF model.

**Figure 12 sensors-20-00132-f012:**
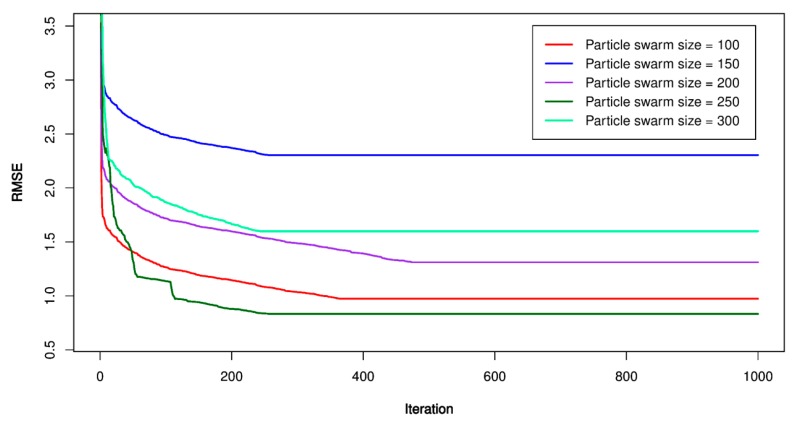
RMSE of the proposed particle swarm optimization (PSO)-SVR-L model.

**Figure 13 sensors-20-00132-f013:**
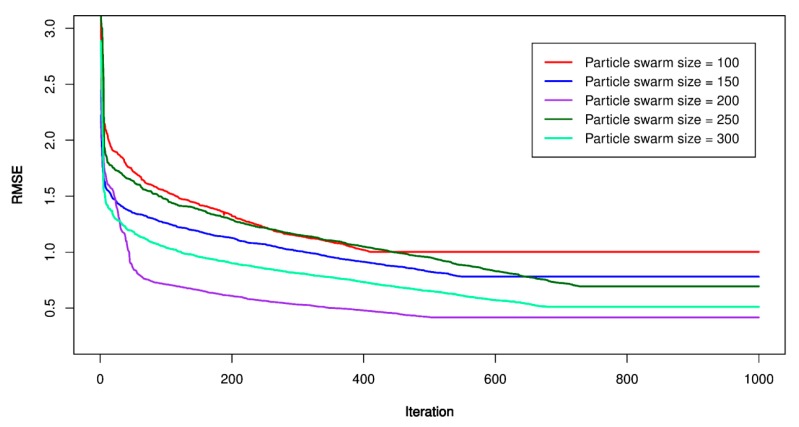
RMSE of the proposed PSO-SVR-P model.

**Figure 14 sensors-20-00132-f014:**
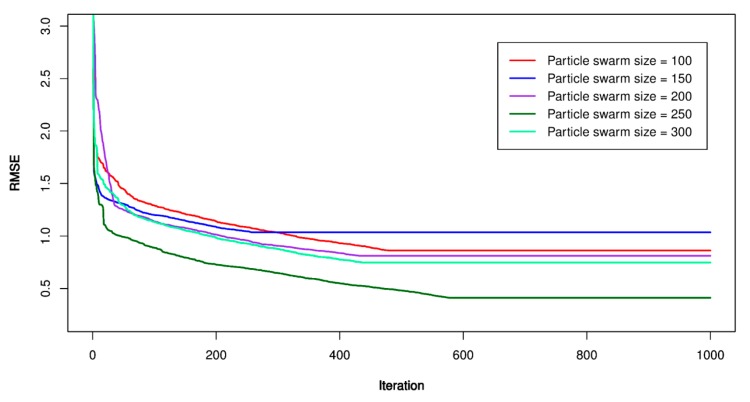
RMSE of the proposed PSO-SVR-RBF model.

**Figure 15 sensors-20-00132-f015:**
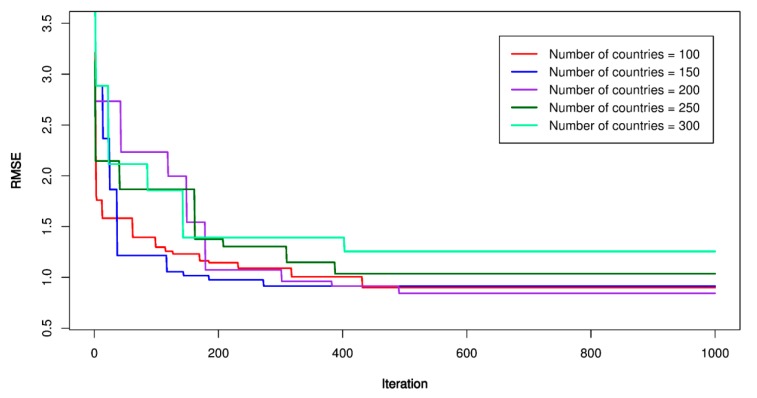
RMSE of the proposed imperialist competitive algorithm (ICA)-SVR-L model.

**Figure 16 sensors-20-00132-f016:**
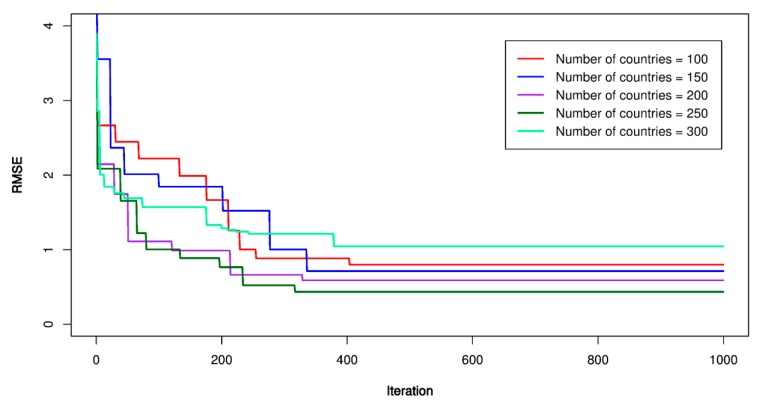
RMSE of the proposed ICA-SVR-P model.

**Figure 17 sensors-20-00132-f017:**
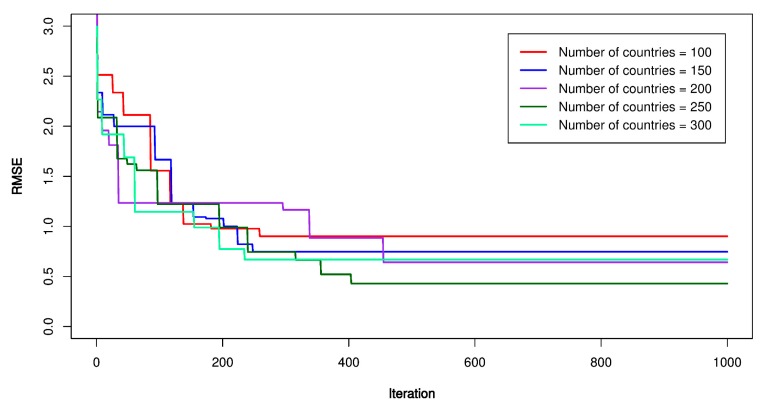
RMSE of the proposed ICA-SVR-RBF model.

**Figure 18 sensors-20-00132-f018:**
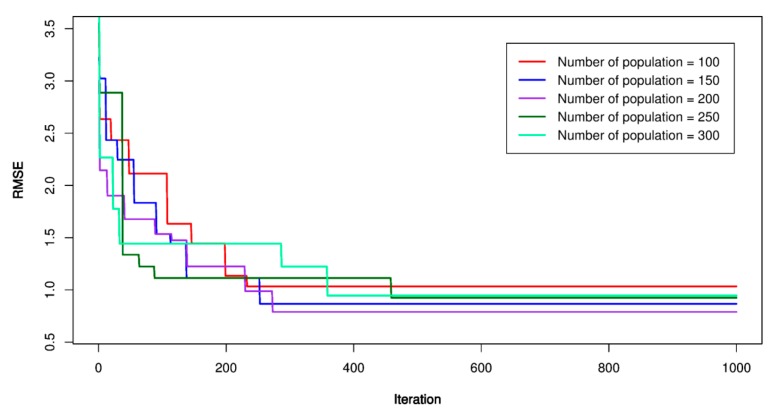
RMSE of the proposed genetic algorithm (GA)-SVR-L model.

**Figure 19 sensors-20-00132-f019:**
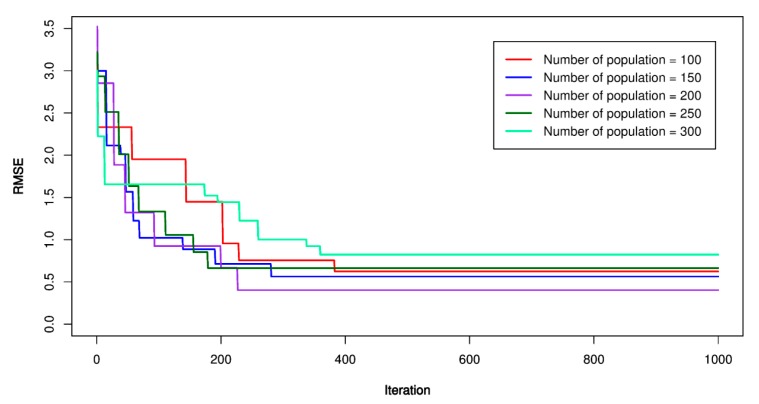
RMSE of the proposed GA-SVR-P model.

**Figure 20 sensors-20-00132-f020:**
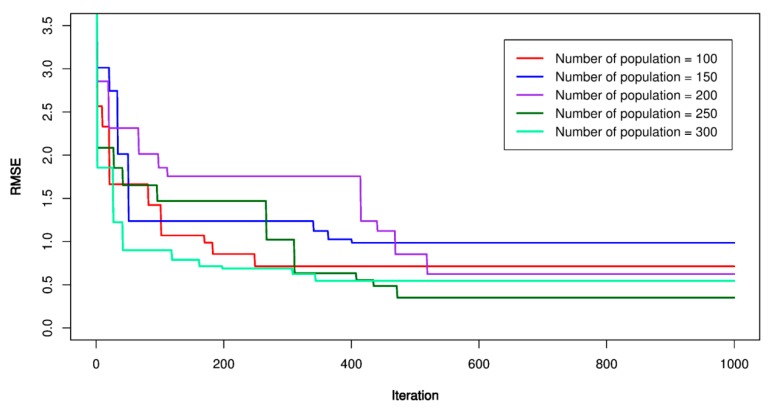
RMSE of the proposed GA-SVR-RBF model.

**Figure 21 sensors-20-00132-f021:**
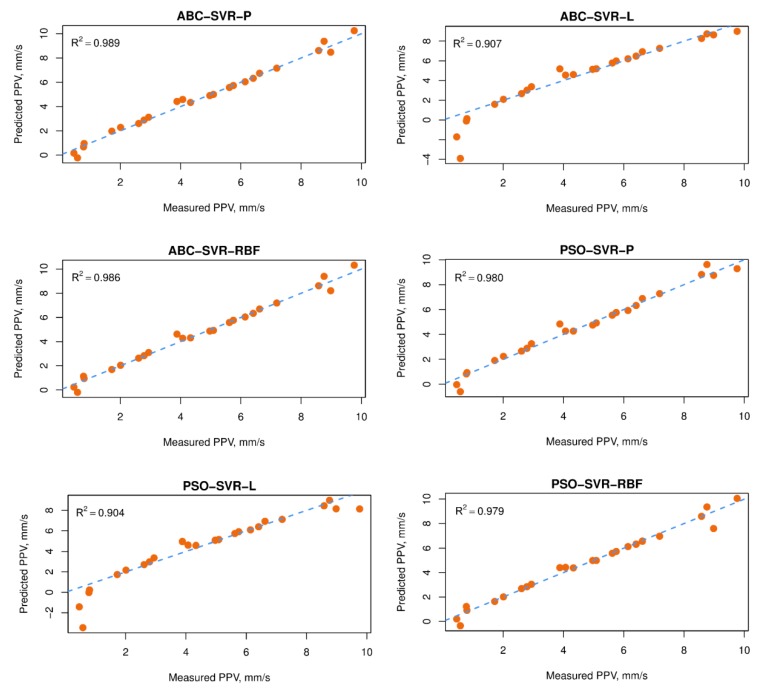
Correlation of measured and estimated PPVs by the 12 proposed hybrid models.

**Figure 22 sensors-20-00132-f022:**
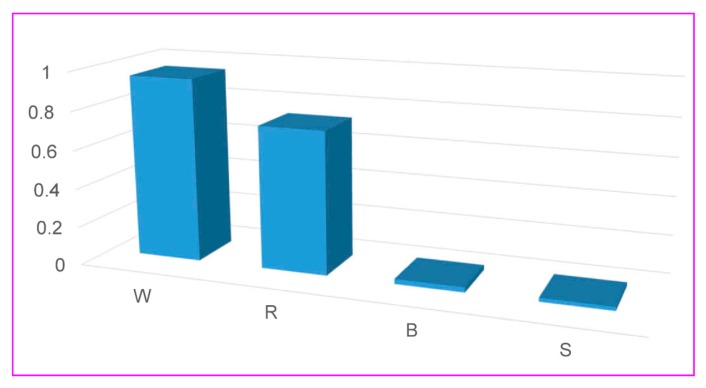
Sensitivity analyses of the influence parameters in this study.

**Table 1 sensors-20-00132-t001:** Some works and their results in predicting blast-induced peak particle velocity (PPV).

Reference	Method	Results
Khandelwal, Singh [34]	ANN	R^2^ = 0.986; MAE = 0.196
Khandelwal et al. [35]	SVM	R^2^ = 0.955; MAE = 0.226
Saadat et al. [36]	ANN-LM	R^2^ = 0.957; MSE = 0.000722
Hajihassani et al. [37]	ICA-ANN	R^2^ = 0.976
Hajihassani et al. [38]	PSO-ANN	R^2^ = 0.89; MSE = 0.038
Amiri et al. [39]	ANN-KNN	R^2^ = 0.88; RMSE = 0.54; VAF = 87.84
Hasanipanah et al. [40]	CART	R^2^ = 0.95; RMSE = 0.17; NS = 0.948
Hasanipanah et al. [41]	PSO-power	R^2^ = 0.938; RMSE = 0.24; VARE = 0.13; NS = 0.94
Taheri et al. [42]	ABC-ANN	R^2^ = 0.92; RMSE = 0.22; MAPE = 4.26
Faradonbeh, Monjezi [43]	GEP-COA	R^2^ = 0.874; RMSE = 6.732; MAE = 5.164
Behzadafshar et al. [44]	ICA-linear	R^2^ = 0.939; RMSE = 0.320; VAF = 92.18%; MBE = 0.22; MAPE = 0.038
Tian et al. [45]	GA-power	R^2^ = 0.977; RMSE = 0.285
Hasanipanah et al. [46]	FS-ICA	R^2^ = 0.942; RMSE = 0.22; VAF = 94.2%
Nguyen et al. [12]	HKM-ANN	R^2^ = 0.983; RMSE = 0.554; VAF = 97.488%
Nguyen et al. [11]	HKM-CA	R^2^ = 0.995; RMSE = 0.475; MAE = 0.373
Zhang et al. [8]	PSO-XGBoost	R^2^ = 0.968; RMSE = 0.583; MAE = 0.346, VAF = 96.083

**Table 2 sensors-20-00132-t002:** Support vector regression (SVR’s) parameters based on the kernel functions used.

Kernel Function	*C*	*μ*	*κ*	*σ*
L	x	-	-	-
P	x	x	x	-
RBF	x	-	-	x

**Table 3 sensors-20-00132-t003:** Summary of PPV database in this study.

Parameter	Min.	Mean	Max.	Standard Deviation
**W**	39.200	54.620	77.900	6.846
**R**	100.000	202.800	380.000	55.751
**B**	2.400	3.312	4.500	0.437
**S**	3.000	3.302	3.600	0.208
**PPV**	0.300	4.804	15.170	2.928

**Table 4 sensors-20-00132-t004:** Optimal values of the artificial bee colony (ABC)-SVR model for predicting PPV.

Model	*C*	*μ*	*κ*	*σ*
**ABC-SVR-L**	0.544	-	-	
**ABC-SVR-P**	0.146	0.729	2	
**ABC-SVR-RBF**	70.067	-	-	0.016

**Table 5 sensors-20-00132-t005:** Optimal parameters of the PSO-SVR models.

Model	*C*	*μ*	*κ*	*σ*
**PSO-SVR-L**	0.119	-	-	
**PSO-SVR-P**	4.995	0.022	2	
**PSO-SVR-RBF**	40.901	-	-	0.036

**Table 6 sensors-20-00132-t006:** Obtained values of the SVR models by imperialist competitive algorithm (ICA) optimization.

Model	*C*	*μ*	*κ*	*σ*
ICA-SVR-L	0.101	-	-	
ICA-SVR-P	128.596	0.002	3	
ICA-SVR-RBF	2.461	-	-	0.079

**Table 7 sensors-20-00132-t007:** Genetic algorithm (GA)-SVR models with the optimal parameters.

Model	*C*	*μ*	*κ*	*σ*
GA-SVR-L	0.178	-	-	
GA-SVR-P	12.730	0.018	2	
GA-SVR-RBF	7.938	-	-	0.030

**Table 8 sensors-20-00132-t008:** Statistical criteria of the PSO-SVR, ICA-SVR, ABC-SVR, and GA-SVR models.

Model	Training Dataset						Testing Dataset						Total Rank
	RMSE	R2	MAE	Rank for RMSE	Rank for R2	Rank for MAE	RMSE	R2	MAE	Rank for RMSE	Rank for R2	Rank for MAE	
ABC-SVR-P	0.425	0.977	0.265	7	9	8	0.317	0.989	0.226	11	11	9	55
ABC-SVR-L	0.754	0.943	0.493	4	3	4	1.107	0.907	0.578	1	3	2	17
ABC-SVR-RBF	0.362	0.981	0.227	11	11	12	0.345	0.986	0.222	9	9	10	62
PSO-SVR-P	0.417	0.976	0.276	8	7	6	0.413	0.980	0.286	6	7	5	39
PSO-SVR-L	0.833	0.941	0.526	2	2	2	1.044	0.904	0.574	4	2	4	16
PSO-SVR-RBF	0.411	0.978	0.256	9	10	10	0.410	0.979	0.245	7	6	8	50
ICA-SVR-P	0.434	0.974	0.286	5	5	5	0.471	0.977	0.272	5	5	6	31
ICA-SVR-L	0.843	0.940	0.530	1	1	1	1.045	0.901	0.580	3	1	1	8
ICA-SVR-RBF	0.428	0.975	0.274	6	6	7	0.335	0.987	0.205	10	10	11	50
GA-SVR-P	0.403	0.976	0.264	10	7	9	0.379	0.983	0.263	8	8	7	49
GA-SVR-L	0.789	0.945	0.510	3	4	3	1.090	0.907	0.577	2	3	3	18
GA-SVR-RBF	0.351	0.983	0.238	12	12	11	0.267	0.991	0.182	12	12	12	71

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
