# Peer review of "Predicting Blast-Induced Ground Vibration in Open-Pit Mines Using Vibration Sensors and Support Vector Regression-Based Optimization Algorithms"

_sensors, 2019, doi:10.3390/s20010132_

Round 1

Reviewer 1 Report

In this paper, the authors used several evolutionary algorithms to optimize a support vector regression (SVR) model to select the best technique for PPV estimation. The paper makes a comprehensive study on SVR behavior when combined with evolutionary algorithms to enhance their performance.

The proposed methodology is clear and described in adequate detail. The figures are adequate and the experimental results are acceptable. In general, the scientific quality of the paper and its relevance to the field of computing are good.

Reviewer 2 Report

You need to be careful of your overuse of commas in some sentences as these change the meaning of the sentences

Line 41 - Among the activities conducted in opencast mines, blasting is a necessary step[,] which leads to serious environmental impacts, such as air and ground vibrations, flying rock, noise pollution, back‐break, and release of gases [1]. …Delete comma after step as not needed and makes sentence incorrect.

Line 43-44 you mention ground vibration is the most dangerous phenomenon – most dangerous to what? A human walking past the site would probably be more affected by flying rock or dangerous gases. Presumably buildings would be most affected in the long term by vibrations. In fact the focus on buildings emerges later but you need to qualify this danger here.

Line 45 – it might pay to define for the reader (who is probably in AI not physical mining) what bench blasting is.

Line 60 – wording is awkward. “Regarding ground vibration prediction, Saadat et al. [23] developed...” avoid starting sentences with Regarding instead reorganize the sentence to … “Saadat et al. [23] developed a new computation model based …. to predict ground vibrations measured as PPV”.

From lines 58-76 a listing of approaches to predict PPV are presented – this is duplicated in Table 1. What I would like to see here rather than the listing is a synthesis of the results of the table rather than a descriptive listing of the methods used. Synthesis for example would for example summarize what gave the best prediction? What are the limitations of these prior works? What is the most common approach e.g. an ANN based approach? In fact some of this is given in the next paragraph – I would retain the paragraph starting 77 and add to the discussion here and remove the content from 58-76 as it does not add much to this lengthy paper.

Line 78 – “However, they were not applied in all areas”. I am not sure what you mean by this please rewrite this sentence as it reads the PPV prediction models were not applied in all areas and I am not sure what these areas you refer to are.

A better justification for the methods explored in this paper is required beyond – “no-one has tried yet”. For example, what about the nature of the PPV data lends itself to the chosen methods. Was a method chosen as a benchmark method which would allow at least some level of comparison of the results of others.

Line 219  missing opening bracket on equation number.

Line 231 “For each best hybrid model obtained, the RMSE value is lowest, respectively to the best‐obtained SVR’s parameters.” This sentence is not complete, please rewrite. The content between the commas should be able to be removed and the sentence still make sense.

Line 255 “The area of the mine is ~ 86 ha with the production ~ 6 Mt/yr.”? Do you mean The mine covers an area of  ~ 86 ha and has an annual production of ~ 6 Mt/yr. (I am not sure what the Mt unit is? Please clarify this unit is it meant to be meters?)

Line 283 – which handheld GPS was used and what is its accuracy in terms of measuring monitoring distance? Detailing this is important.

Line 251 the yi parameter is not in line with the sentence text

Section 3 heading is not at correct tab distance

The paper is overly long and this will put off readers. In particular in Section 2 tediously repeats standard and well know data mining techniques in detail – these methods are not new by any means. Figures 1 and 2 are also not necessary. The level of detail provided is not necessary, all that is required is the reference to the method. For example for PSO the reader simply needs to be referred to Kennedy’s paper they do not need a repetition of the algorithm reported in the paper.

Details are only needed when you have modified an approach/algorithm and the details are not present in the literature for that implementation or you have created a unique hybrid approach that warrants details. When using/implementing a standard method the details of parameters (and if necessary tuning approach) is all that is needed.

A geophone sensor is used – did you build this sensor or was it purchased – if purchased who from? How was the sensor calibrated prior to measurement? Please detail this from Fig 8 it appears that the sensor was an Instantel sensor? This should be appropriately documented along with specific details of the model used.

Section 4.1 line 294 needs a reference to the data sets used in the USBM model (and a link to that data set for others to be able to duplicate your work).  Details of the test/train split of data is required to help the reader.

Reviewer 3 Report

In this paper, the authors attempted to predict blast-induced ground vibration based on the support vector regression and some meta-heuristic algorithms. Vibration sensors were used for measuring the intensity of ground vibration with high reliability. 12 hybrid models were developed for predicting blast-induced ground vibration with high accuracy, and a comprehensive assessment of these 12 models was conducted. It is a very interesting work and falls into the theme of the journal. The paper was well-organized and written, easy for understanding and following. Some minor revisions are needed before acceptance:
1. The x-axis in figure 10 needs to be revised and addressed.
2. Line 305: Figures 11-13
3. How to select the training and testing datasets for model development and testing?
4. How to avoid over-fitting or under-fitting?
5. Line 374: Figures 20-22.

6. Some relevant references related to blast vibration and SVM are recommended to enhance the descript of Introduction. For exmaple,

Feasibility of the indirect determination of blast-induced rock movement based on three new hybrid intelligent models

Computational intelligence model for estimating intensity of blast-induced ground vibration in a mine based on imperialist competitive and extreme gradient boosting algorithms

Utilizing gradient boosted machine for the prediction of damage to residential structures owing to blasting vibrations of open pit mining
